# Circadian Regulation for Optimizing Sport and Exercise Performance

**DOI:** 10.3390/clockssleep7020018

**Published:** 2025-04-07

**Authors:** Garrett R. Augsburger, Eric J. Sobolewski, Guillermo Escalante, Austin J. Graybeal

**Affiliations:** 1Department of Kinesiology, Texas Christian University, Fort Worth, TX 76129, USA; g.r.augsburger@tcu.edu; 2Department of Health Sciences, Furman University, Greenville, SC 29613, USA; eric.sobolewski@furman.edu; 3Department of Allied Health Professions, Liberty University, Lynchburg, VA 24515, USA; 4Department of Kinesiology, California State University San Bernardino, San Bernardino, CA 92407, USA; gescalan@csusb.edu; 5School of Kinesiology and Nutrition, University of Southern Mississippi, Hattiesburg, MS 39402, USA

**Keywords:** circadian rhythm, diurnal, time of day, muscle performance, time zone, chronotype

## Abstract

This narrative review explores the intricate relationship between circadian regulation and exercise performance, emphasizing the importance of aligning training strategies with the body’s natural physiological fluctuations. The three key mechanisms investigated are temperature, hormonal fluctuations, and diurnal chronotype—an individuals’ exhibition of enhanced physical or cognitive performance at specific times of day. Core body temperature variations impact exercise performance, suggesting strategic workout timing and intensity adjustments. Hormonal patterns (i.e., insulin, cortisol, testosterone) influence energy metabolism and muscle growth, informing tailored training plans. Diurnal chronotypes significantly affect performance, advocating for personalized training sessions based on individual preferences and entrained awakening. Integrating circadian mechanisms into training offers strategic advantages, guiding practitioners to design effective, personalized regimens, though we acknowledge relevant challenges and the need for further research.

## 1. Introduction

Optimizing sport and exercise performance requires adapting training to time-of-day-dependent fluctuations in physiological regulation [1]. These daily fluctuations, known as circadian rhythms, are driven by internal physiological changes over a 24 h cycle and represent the outputs from the circadian time systems. Functions of the neuromuscular system, such as muscle strength, activation, force production, and steadiness, are regulated by these rhythms, not just the immediate environmental stimuli [2]. While the complexities of circadian regulation have yet to be fully understood, they encompass the physical, mental, and behavioral changes that follow the diurnal sleep–wake cycle [3]. The influence of circadian regulation on performance involves factors like core body temperature, neuromuscular excitation, and hormonal fluctuations [2,4,5,6]. Moreover, it has been hypothesized that there are potential genetic factors that affect how circadian systems function [7].

Given that sporting events occur across different time zones and times of day, adapting training strategies to align with circadian regulation is essential. For elite-level athletes desiring a competitive edge, accounting for circadian regulation could lead to marginal, yet meaningful advantages over their competition. Practitioners and coaches need to increasingly consider the impact of circadian regulation on physical performance parameters to create training programs that better facilitate within-session performance and avoid circadian-driven deficits. Therefore, the purpose of this article is to illuminate the specific time-of-day effects on sport and exercise performance and discuss their underlying mechanisms. Additionally, this article will explore strategies to enhance athletic and muscular performance by leveraging these time-of-day effects for athletes seeking a competitive advantage.

## 2. Physiological Parameters Modulated by Circadian Rhythms

At the cellular level, circadian time systems rely on the set of interconnected transcription factors, named clock proteins, that form an oscillating transcription/translation feedback loop spanning a period of 24 h [7]. This tissue-specific loop orchestrates circadian gene expression, which facilitates the circadian regulation of various cell functions [7]. In mammals, the suprachiasmatic nucleus (SCN) of the hypothalamus originates our circadian rhythm, acting as a central pacemaker that entrains peripheral oscillators through neural, humoral, and hormonal signals [7]. Previous studies have highlighted the tissue-specific nature of this mechanism, revealing that skeletal muscle possesses its own inherent circadian clock output—a phenomenon observed in various tissue types [7]. It has been established that receiving solar light (i.e., light from the sun) into the optic channels drives the modulation of these internal clocks, although other stimuli, such as ingestion of food and physical exercise, modify the activity of the clock genes at the central and peripheral levels.

Previous literature has shown that circadian regulation both positively and negatively affect neural excitation and motor unit activation during muscle contractions [8,9]. Differing times of day exhibit altered rates of force development (RFD) and electromyographic (EMG) rise during rapid contractions as indicated by force steadiness measurements [2]. These metrics provide insights into the magnitude of motor unit activation, which vary throughout the day. Circadian regulation may also impact control of the contractile properties at the muscle fiber level. Studies have shown time-dependent variations in muscle contractility, reporting that muscle performance peaks in the afternoon [10,11]. While initial theories have attributed fluctuations in muscular performance to changes in neural drive, the predominating theories currently focus on peripheral changes in the skeletal muscle, such as changes in contractile properties, neural conduction velocity, substrate utilization, and connective tissue extensibility that are partially independent of central mechanisms [2,12].

Time-of-day dependent variations in maximal performance during waking hours has been documented almost exclusively in laboratory settings [2,7], but more recent evidence demonstrates that these variations can occur during competitive sporting events [13]. Performance in maximal exercise bouts follow a time-of-day dependent pattern with peaks occurring in the afternoon and evening [14,15], which has been shown across numerous exercise modalities and sports [16,17]. While circadian-dependent mechanisms have considerable effects on various markers of physical performance, their effect on muscular strength is of particular interest to coaches, athletes, and practitioners [18].

Evidence from laboratory and applied settings suggests that maximal strength peaks in the evening (around 16:00–20:00 h) and is lowest in the morning (around 06:00–10:00 h) [2,10,11,19]. It has also been hypothesized that circadian clock genes contribute to these fluctuations, although the research in this area is still evolving [7]. Because the morning hours often represent the most available training window for athletes, coaches and other practitioners should be aware of the potential for time-dependent training deficits and seek to employ mitigation strategies.

Prior research suggests that the influence of circadian regulation on neuromuscular function and muscle strength may be sex-specific, where recent studies indicate that males exhibit greater time-dependent force variability than females during isometric contractions [2]. While a correlation between circadian fluctuations in neuromuscular performance and the circadian regulation of gonadal steroid concentrations may provide a potential explanation (see Section 3.2), most research has focused exclusively on males, and there is a paucity of literature regarding whole-day variation in neuromuscular function in females [12,20].

Overall, circadian regulation influences various physiological factors related to physical performance, including muscle contractility, neural activation, and temperature regulation, highlighting the complex interplay between the body’s internal clock and its impact on exercise capacity throughout the day.

## 3. Mechanisms Leading to Diurnal Variations in Performance

### 3.1. Mechanism I: Temperature

The connection between physical performance and natural fluctuations in core body temperature has been extensively studied, revealing a compelling association. Research indicates that both core body temperature and specific parameters of physical performance tend to reach their peak levels in the early evening hours [15,16]. This diurnal variation in performance appears to be linked to passive heating of muscles within a physiological range, which enhances the contractile properties of muscle fibers. These improvements are believed to stem from variations in intracellular calcium kinetics and excitation–contraction coupling mechanisms. Intracellular calcium kinetics involves the regulation of calcium ions within muscle cells and excitation–contraction coupling is the process by which an electrical stimulus triggers muscle contraction. At elevated temperatures (such as the early evening hours), these processes become more efficient, improving muscle strength and power [21,22,23].

Unintentional fluctuations in body temperature occur throughout the day due to routine activities and factors unrelated to exercise. This passive warm-up effect explains why peak physical performance is commonly observed in the late afternoon and early evening rather than the morning hours [24]. Elite athletes showcase elevated performance levels during the evening hours, highlighting the relevance of diurnal fluctuations in optimizing athletic performance outcomes [25]. When passive warming is infeasible, such as during morning training, exposure to a warm environment (28–29.5 °C) for just an hour mitigates the typical shifts in physical performance observed during diurnal variations under neutral climate conditions [23,26]. This highlights the potential for intentional environmental interventions or properly implemented warm-up protocols to modulate body temperature and, consequently, optimize exercise performance throughout the day. Additionally, athletes who begin activity earlier in the day achieve a peak core body temperature sooner, better preparing them for physical performance. Coaches and practitioners should consider strategies to increase body temperature prior to exercise onset, especially in low environmental temperatures (e.g., winter training), as low temperatures negatively impact physical performance parameters such as rate coding and neural conduction velocities. [19,26,27].

#### 3.1.1. Timing of Training Sessions

Understanding the diurnal variations in core body temperature and their impact on physical performance is vital for timing optimal training sessions. Table 1 displays a compilation of studies that have investigated the effects of circadian-driven temperature fluctuations on changes in muscular/athletic performance. Overall, athletes need to consider scheduling their training sessions during these times if congruency between training and competition is the desired outcome [28,29]. However, barriers such as schedule availability remain a significant obstacle for those trying to plan training around circadian dependent temperature spikes. Thus, coaches and other practitioners looking to leverage the effects of ambulated body-temperature must schedule training sessions within peak performance hours, or schedule training with enough warm-up time to offset any body temperature-related detriments in physical performance.

#### 3.1.2. Active Muscle Warming and Training Periodization

Passive heating of muscles enhances physical performance; however, actively warming muscles through proper warm-up protocols achieves similar effects [26,35]. This approach proactively prepares muscles for physical exertion by engaging in effective warm-up routines that gradually increase body temperature, enhance blood flow, and optimize the contractile properties of muscle fibers. Therefore, coaches and practitioners should emphasize dynamic warm-ups tailored to specific activities. Countering morning performance deficits due to temperature has implications for athletes who have early training or competition sessions [36]. To counteract these deficits, practitioners can also employ strategies such as utilizing thermal outerwear or hot water immersion [37]. While these strategies may help alleviate the negative effects of lower temperatures, they may not always be practical in certain training environments. Thus, implementing a dynamic warm-up is often a more feasible option.

Understanding the relationship between temperature, performance, and training time can guide training periodization, prompting coaches and athletes to adjust training intensity, duration, and type based on diurnal body temperature variations [38,39]. Implementing warm-up protocols that elevate heart rate and core body temperature earlier and for extended periods of time will help offset performance detriments, leading to greater training-induced adaptations and overall performance. Tailoring training routines to body temperature fluctuations, including passive and active warming, session timing, and environmental temperature, optimizes physical performance. [26,36].

### 3.2. Mechanism II: Hormonal Fluctuations

It has established that hormones like insulin, cortisol and testosterone follow distinct diurnal patterns [40,41,42]. Prior investigations have shown that performance improvements are associated with lower insulin and cortisol levels; each of which exhibiting distinct time-of-day circadian fluctuations [8,43,44]. The body’s sensitivity to insulin, for example, is notably higher in the morning and lower in the evening—creating a situation in which typical detriments in early-morning performance are mitigated by more efficient glucose uptake in working tissues [45]. As such, if glucose is used more effectively during periods in which eating opportunities may be limited prior to training (i.e., early morning), morning performance deficits might be less severe. However, it is unknown if circadian-dependent increases in insulin sensitivity are enough to overcome the limitations of low absolute energy intake and lower core body temperature; thus, future research is warranted.

Cortisol also follows a diurnal pattern, where it peaks in the early morning shortly after awakening—otherwise known as the cortisol awakening response [46]. Following morning elevations, cortisol concentrations gradually decline over a 24 h period, reaching their lowest point in the late evening and early night. Despite the negative connotations associated with circulating cortisol, from a performance perspective, moderate increases in cortisol can have positive effects. Cortisol assists in mobilizing energy stores, increasing alertness, and supporting the body’s response to stressors, including exercise. At modest concentrations, cortisol has been shown to contribute to improved performance by converting stored glycogen into glucose, which is subsequently used by skeletal muscle to fuel exhaustive exercise, mitigating performance detriments experienced during morning training [44,47]. It should be noted that this is the active mechanism during fasting or when food is not readily available and thus, it is not an ideal training scenario as peak performance is usually achieved in a fed state [48].

Testosterone, an important sex hormone in both males and females, is known to play a role in muscle growth, bone health, and overall vitality. Like cortisol, testosterone levels also follow a circadian pattern, reaching peak levels in the early morning, declining throughout the day, and reaching their lowest point in the evening [40,41,46]. Given that testosterone is associated with muscle protein synthesis (MPS), a key process in the development and reparation of muscle tissue, higher testosterone concentrations are critical for recovery and for achieving positive adaptations to training [47]. Though literature remains varied, some research has suggested that training during periods of elevated testosterone enhances the body’s ability to acutely recover from exercise and promotes MPS during rest, which is crucial for overall performance [47,49,50]. This is an important consideration as increasing recovery rates from training may allow for increased training volume and frequency, thus providing a stimuli for enhanced long-term performance.

The intricate relationship between circadian fluctuations in insulin, cortisol, and testosterone levels and their impact on performance is highly contextual. While the morning surge in insulin sensitivity presents a potential mitigator for early-morning performance deficits, it remains uncertain whether this alone counterbalances the constraints imposed by lower energy intake and core body temperature. Similarly, despite cortisol’s diurnal peak in the morning, its role in mobilizing energy stores may contribute positively to performance, particularly during morning training. Furthermore, the early morning peak in testosterone levels in the underscores its significance for muscle growth and recovery, presenting an optimal window for enhancing training adaptations. Nevertheless, the broader context indicates that these hormonal components function more as moderators than as direct performance enhancers, highlighting the need for comprehensive consideration of the various circadian-related factors that influence overall performance outcomes. Further research is warranted to unravel the complex interplay between these factors, and provide a more holistic understanding of how these hormonal patterns contribute to the intricacies of performance dynamics throughout the day.

#### 3.2.1. Melatonin and Sleep

Melatonin, a hormone produced in the pineal gland that follows circadian pulsing patterns, exerts widespread effects on the body as a neuroendocrine transmitter and plays a crucial role in regulating behavior and physiology through its rhythmic synthesis [43,51]. The synthesis of melatonin is directly regulated by the SCN, and it involves a sympathetic pathway that releases noradrenaline to the pinealocytes during the night to facilitate the synchronization of circadian rhythms across various biological functions [51,52]. Its influence as a sleep-inducing agent is well-documented, which serves as a reliable predictor of sleep onset [51]. Acting as a natural rhythm synchronizer, melatonin aids in promoting sleep by inducing vasodilation, leading to a decrease in core body temperature, blood pressure, and other muscle specific mechanisms such as the inhibition of calcium release from the sarcoplasmic reticulum [51,52].

Prioritizing good sleep hygiene, including creating a dark and cool sleep environment, supports better sleep quality [53,54]. Because melatonin also helps lower core body temperature [53], it is important to end training prior to the onset of melatonin release, as the sleep promoting effects leads to diminished performance and wasted time [51,55]. For athletes who travel across time zones, understanding melatonin’s role in circadian rhythm synchronization can guide strategies for minimizing jet lag. Gradual adjustment to new time zones, exposure to natural light, and proper sleep planning will help mitigate disruptions in performance [56,57]. Adjusting training for sport around occupational constraints may also force individuals into irregular work schedules that disrupt their circadian rhythms. These individuals benefit from establishing a regular sleep routine, even if their work hours vary, to promote better sleep quality and recovery from training [58]. Aspiring athletes with shift work schedules who need to sleep during the day should be aware of melatonin’s role in inducing sleep. Understanding this will help them optimize their sleep environment and habits to support recovery between shifts. Engaging in vigorous exercise right before attempting daytime sleep disrupts sleep quality, making it important for these individuals to plan workouts accordingly.

Generally, engaging in vigorous exercise prior to sleep disrupts sleep quality. Exercise should be planned according to an individual’s sleeping schedule, ideally terminating vigorous activity at least 2–3 h before an individual’s desired bedtime [59]. This timeframe allows core body temperature to decrease and melatonin levels to rise, both of which are critical for initiating restful sleep. Exercising too close to bedtime can delay these physiological processes, leading to difficulty falling asleep and reduced sleep efficiency [60]. However, the optimal separation time may vary based on individual factors such as chronotype, exercise intensity, and personal tolerance. For athletes or individuals with irregular schedules, tailoring this guideline to their specific needs can further enhance recovery and performance.

#### 3.2.2. Hormonal Signals

The SCN, as well as the nuclei in brain cells of the ascending and descending pathways of the central nervous system, all bear estrogen and androgen receptors [61] that allow for bidirectional communication [45]. As circadian regulation influences many aspects of human physiology and behavior, alterations in regulatory control mechanisms due to the presence of estrogen and androgen receptors could lead to changes in performance, suggesting that individual hormonal profiles may have independent circadian clocks that sync with central circadian rhythms through the stimulation of melatonin and other circadian modulatory hormones. Specifically, androgens and estrogens exert their effects on various target tissues, including the brain, by entraining and sychronizing peripherial and central circadian systems that drive the alterations in diurnal performance [45,62,63,64,65]. Consquently, the stimulation of these hormones alters the timing of circadian regulation, which affects the peak of certain physiological processes during the day, including hormone secretion, body temperature, and cognitive functions [20,66]. Therefore, performance in physical exercise may be influenced by the timing of peripherial hormone release and circadian regulation [29].

#### 3.2.3. Training Implications of Hormonal Signaling Mechanisms

Determining an individual athlete’s hormonal profile as it pertains to daily fluctuations requires specific medical tests, such as the collection of blood or saliva at various times of the day. While understanding an athlete’s hormonal profile remains valuable in optimizing training plans and workout timing, obtaining time-specific hormonal profiles is challenging and impractical. As such, improving overall knowledge of the general influence of diurnal hormonal fluctuation will allow coaches and trainers to modulate training intensity and volume in alignment with hormonal fluctuations, potentially optimizing adaptations and recovery. Practitioners also need to be aware of the significant sex differences in diurnal-related hormonal profiles, where males and females have different baseline sex hormone concentrations with distinct fluctuation patterns [67,68].

Given the distinct circadian fluctuations in hormones like melatonin, insulin, cortisol, and free testosterone (FT), it is beneficial to consider hormone levels when scheduling workouts. For instance, higher levels of testosterone and other hormones, like growth hormone (GH), are associated with improved performance [69]. Numerous studies have explored the circadian regulation of testosterone levels and have reported that peak concentrations typically range from 750 to 800 ng/dL (26 to 28 nmol/L) and occur around 06:00 h to 08:00 h; however, nadir (lowest point) concentrations of ~500 ng/mL (17 nmol/L), are observed during the 18:00 h to 20:00 h timeframe [42,70,71]. Therefore, planning high-intensity training sessions during periods when these hormones are naturally higher can enhance exercise performance [72].

The fluctuating levels of insulin and glucose suggest that nutrient timing plays a role in diurnal performance optimization. Carbohydrate consumption could be strategically timed around periods of increased insulin sensitivity, potentially maximizing glucose uptake during workouts, and mitigating the diminished exercise performance observed in the morning. Given that many carbohydrate-rich foods are often “ready-to-eat” (i.e., sports drinks, gels, energy bars, etc.) and do not require significant preparation time, early-morning carbohydrate consumption may be a feasible eating option that is well-aligned with time-dependent shifts in insulin sensitivity.

In summary, hormonal fluctuations, as well as melatonin’s role in circadian regulation, have profound implications for optimizing sport and exercise training [4]. Therefore, coaches and trainers should, when possible, assess an individual athlete’s hormonal profile through blood testing, as this information helps practitioners design more effective and personalized training plans that align with the athlete’s natural biological rhythms. When hormonal testing is not possible, general knowledge of diurnal fluctuations, sex differences, and the interaction between hormones and circadian regulation will assist in optimizing training adaptations and recovery. By tailoring workout timing to individual hormonal profiles and sleep hygiene needs, practitioners and athletes can optimize training plans, lifestyle, and enhance performance outcomes. From a practical standpoint, however, it should be noted that not all these factors can be feasibly prioritized, and consistent training remains imperative.

### 3.3. Mechanism III: Diurnal Chronotype

The circadian rhythm in humans manifests in a complex phenotype derived from multiple genetic factors that define one’s chronotype [7,48]. Diurnal chronotype, defined as an individual’s diurnal preference (i.e., whether they exhibit enhanced physical or cognitive performance in the morning or in the evening) is considered to exist across a spectrum rather than a dichotomous classification [73]. For instance, certain individuals trend heavily toward a ‘strong morning’ or ‘strong evening’ chronotype, while others have chronotypes that exist in-between [24]. Regarding performance, it has been suggested that an individual’s skeletal muscle-specific circadian clock is a mediator of strength and physical performance [7].

Numerous morningness–eveningness questionnaires (MEQ) exist to determine ones’ diurnal chronotype and have utility for practitioners seeking to enhance exercise prescription. Notably, the instrument developed by Horne and Ostberg (1976) [74] is considered the most accurate depiction of diurnal preference in the general population [24]. Evidence has shown that an individual’s diurnal chronotype plays a major role in determining their performance during differing times of the day [24]. Some researchers have also pointed out that ‘time-of-day’ is an exogenous effect and therefore, reflects only part of the picture when analyzing an individual’s circadian-related physiology and performance. Analyzing results based on diurnal preference, and as a function of time, is beneficial in that it considers the differences in biological days [13,24].

Traveling across (multiple) times zones for both athletes and the general population, desynchronizes the internal body clock. This desynchronization results from sleep disruptions and jet lag commonly experienced during travel. This is of particular concern among high level athletes, as this leads to misalignment between the body’s internal clock and the actual chronological time prior to competition [56,57]. Increased knowledge of the impact of circadian disruptions leads to more effective intervention designs that minimize sleep disturbances and stabilizes the circadian rhythms [73]. Coaches and practitioners should utilize the previously mentioned MEQ’s to assess an athlete’s diurnal chronotype.

Chronotype significantly influences diurnal patterns of hormonal fluctuations and core body temperature regulation. Morning chronotypes typically exhibit earlier peaks in cortisol and melatonin levels, aligning with a preference for earlier activity and sleep–wake cycles. Conversely, evening chronotypes tend to have delayed peaks in these hormonal rhythms, coinciding with later peak alertness and physical performance [75]. Similarly, core body temperature fluctuations also vary by chronotype; morning types reach their peak daily temperature earlier in the afternoon, while evening types may sustain elevated temperatures later into the evening [76]. Because physiological readiness varies throughout the day and is tailored to a chronotype, these variations are crucial for optimizing performance. Understanding these chronotype-specific hormonal and thermoregulatory patterns allows practitioners to better align training protocols with an athlete’s biological rhythms, enhancing performance and recovery.

An athlete’s performance is influenced not only by their diurnal preference but also by the time since entrained awakening. While there is no conventional definition for circadian entrainment, this article will define it as follows: Circadian entrainment facilitates the synchronization of the internal clock’s period with that of the external cues, such as awakening with the sun and becoming tired at sunset. This entrainment involves aligning one’s circadian rhythm to external cues in the environment that might lead to enhanced performance at specific times of day. External factors such as food timing and light exposure also play important roles in this process. For instance, the timing of meals can influence the synchronization of the internal clock, as eating at specific times can either advance or delay the circadian rhythm [77]. Similarly, melatonin supplementation, when used strategically, can help regulate the sleep–wake cycle, especially in athletes adjusting to irregular schedules or time zone changes [53]. This type of training, like any other form of physiological training, involves responding to a stimulus to augment physiological variables, in this case, the body’s ability to perform work at specific times of day. Circadian entrainment is useful when an athlete has competitions that lie outside of a ‘normal’ societal sleep–wake cycles, and thus, they must ‘train’ their circadian system to perform outside of these parameters [56,57]. As such, recognizing that “time-of-day” only partially captures this phenomenon emphasizes the importance of considering an athletes’ internal clock when planning training [24].

#### Training Implications

Detecting an athletes’ biorhythms through MEQ’s, hormone panels, and normal sleep–wake patterns allows practitioners to optimize training through diurnal specific protocol alterations. Biorhythms and diurnal chronotypes are specific to the individual athlete; however, these remain difficult to identify on the individual level. Regardless, coaches and practitioners should be attentive to individual athletes and adapt training to align with the athletes’ personal biorhythms. It has been shown that individuals will perform better at the time-of-day in which they habitually train, otherwise known as diurnal specificity [24,34]. Targeting training in and around the time that an athlete will compete is crucial in providing a competitive advantage. This is critical for athletes competing in sports such as mixed martial arts that are under media pressure to travel long distances and compete during late hours. This irregular schedule disrupts their natural circadian rhythms, making it harder to synchronize their internal clocks with external cues, affecting their performance. Preparing both the physical body and internal clock prior to competition gives athletes an advantage over their counterparts who do not undergo circadian entrainment or train under diurnally specific conditions. Understanding how to detect diurnal chronotype and properly implement diurnal specificity is an important tool for practitioners to implement in exercise prescription [78].

In individual sports, implementing a MEQ to determine an individual’s diurnal preference or chronotype should be included in the intake/screening process as practitioners assess new clients/athletes/patients. For example, the RB-UB chrono analysis MEQ designed by Facer–Childs and Brandstaetter may be better designed to examine sleep/wake-related parameters that affect training, competition, and performance variables in athletes, and could be implemented by coaches to enhance their athletes training protocols [24]. In team sports, where individual chronotypes and hormonal profiles may vary significantly among players, a tailored approach is essential for optimizing performance. Coaches can use chronotype assessments, such as the MEQ, to identify individual preferences and tendencies. Based on these assessments, staggered training sessions or flexible practice schedules may be implemented to align with players’ peak performance windows [78]. When such adjustments are not feasible, strategies such as targeted warm-ups and cognitive stimulation can help synchronize team members’ readiness to perform. Additionally, educating players on managing their sleep–wake cycles and incorporating light exposure interventions can further support synchronization and mitigate circadian mismatches during competition. By integrating these strategies, team-wide performance can be optimized while still accounting for individual variability.

## 4. Conclusions

This narrative review highlights the complex and dynamic relationship between circadian rhythms and athletic performance, emphasizing the importance of aligning training with the body’s natural biological rhythms. Diurnal fluctuations in physiological parameters such as hormonal levels, core body temperature, and individual chronotypes provide valuable insights for optimizing training schedules and recovery strategies. While consistency in training should be prioritized when precise alignment is not possible, considering circadian patterns during program design can offer athletes and coaches a significant competitive advantage and mitigate the risk of performance deficits.

The importance of managing circadian disruptions caused by societal demands, such as travel across time zones and irregular competition schedules, cannot be overstated. Athletes required to perform at non-optimal times due to external pressures may benefit from targeted interventions that stabilize their circadian rhythms and adapt to new schedules more effectively. By understanding and leveraging these insights, coaches, trainers, and practitioners can create strategies that enhance athletic performance even when margins for error are small.

Future research should explore chronotype-specific training protocols, focusing on refining and assessing their effectiveness across various sports and positions. Researchers should also develop practical interventions to mitigate the effects of circadian disruptions, such as travel-induced jet lag, to enhance adaptability and maintain optimal performance.

### Practical Applications

Understanding the insights from circadian rhythm research offers actionable strategies for practitioners aiming to improve athletic outcomes. Using tools like the MEQ and RB-UB chronometric tests to identify an individual’s diurnal preference can help develop personalized training schedules aligned with internal biological clocks. Coaches should consider the following: (1) adjusting training sessions to align with peak performance times, particularly in the afternoon when physiological parameters are optimal, especially in moderate climates; (2) implementing strategies to counteract the challenges of early morning training, such as extended warm-ups and exposure to warm, humid environments; and (3) employing circadian entrainment techniques to help athletes adapt to early or late competition schedules or travel across time zones. While this paper takes a broad approach to the terms “sport” and “exercise performance”, future work should address how specific sports, roles, and positions might uniquely respond to circadian factors. These practical strategies are particularly relevant for elite athletes aiming to maximize their potential through training in alignment with their body’s natural rhythms.

## Figures and Tables

**Table 1 clockssleep-07-00018-t001:** Circadian Temperature Fluctuations and Associated Changes in Various Performance Metrics.

Study	N	Intervention	Performance Differences by Time-of-Day	Peak Performance Time	Improved Performance Outcome
Souissi et al., 2010 [15]	20 ten- and eleven-year-old boys	Acute performance testing at 08:00 h, 14:00 h, and 18:00 h on separate days	Yes	Afternoon and Evening	Handgrip strength, squat jump, five-jump test, Wingate peak and mean power
Giacomoni, Billaut and Falgairette 2006 [30]	12 active males	Acute repeated cycle sprint tests at 08:00–10:00 h and 17:00–19:00 h on separate days	Yes	Morning	Peak efficient torque
Hatfield et al., 2016 [16]	7 active males	Acute performance testing at 04:00 h, 10:00 h, 16:00 h, and 22:00 h on separate days	No	None	None
Chtourou et al., 2015 [1]	31 young active males	14 wk resistance training performed at 07:00–08:00 h or 17:00–18:00 h	Yes	Evening	Squat jump, MVC
Chtourou et al., 2012 [19]	30 young active males	8 wk lower-extremity progressive resistance training performed at 07:00–08:00 h or 17:00–18:00 h	No	None	None
Hammouda et al., 2012 [14]	15 young male athletes	Acute performance testing at 07:00 h and 17:00 h on separate days	Yes	Evening	Wingate peak power
Pallarés et al., 2013 [31]	12 trained swimmers (6 males, 6 female)	Acute performance testing at 10:00 h and 18:00 h on separate days	Yes	Evening	Max bench press, bench press power, CMJ height, 25 m swimming freestyle
Robinson et al., 2013 [32]	10 active males	Acute performance testing at 07:30 h, 17:30 h, and 17:30 after cold-water immersion on separate days	Yes	Evening	Handgrip strength, isometric peak power, isokinetic knee flexion and extension for peak torque and peak power, knee extension for peak torque
Pullinger et al., 2018 [5]	12 trained males	Acute repeated treadmill sprint tests at 07:30 h and 17:30 h, and three “optimal temperature” trials on separate days	Yes	Evening	Distance covered, sprint mean power and velocity
Sedliak et al., 2008 [9]	32 males	Acute performance testing at 07:00–08:00 h, 12:00–13:00 h, 17:00–18:00 h, and 20:30–21:30 h on two consecutive days	Yes	Evening	Squat jump strength and power
Robertson et al., 2018 [33]	30 resistance-trained males	Acute performance testing at 07:30 h and 17:30 h on separate days	Yes	Evening	Bench press and back squat mean force, mean velocity, and time-to-peak velocity
Küüsmaa-Schildt et al., 2017 [18]	51 young men	24 wk combined resistance and endurance training performed in the morning or evening	No	None	None
Zbidi et al., 2016 [34]	20 active men	6 wk MIVCC training of the right elbow joint at 07:00–08:00 h or 17:00–18:00 h	No	None	None

Studies examining time of day changes in various physiological performance variables. CMJ: counter movement jump; MVICC: maximal voluntary isometric co-contraction; MVC: maximal voluntary contraction.

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
