# Peer review of "Circadian Regulation for Optimizing Sport and Exercise Performance"

_2624-5175, 2025, doi:10.3390/clockssleep7020018_

Round 1
Reviewer 1 Report
Comments and Suggestions for Authors
The review by Augsburger et al rise the problematic about time-of/day variations in sport exercise performance. The subject is relevant for a broad audience as there are few evidence on the mechanism by the circadian time system can determine performance. The authors prepared a good table to correlate the time of exercise, recording of core body temperature and performance. Nevertheless, this table is poorly explored in the arguments throughout the review, which shallow touch the mechanism behind the correlation of circadian rhythm and performance. They should also bring a table like that to correlate other circadian rhythm such as cortisol, testosterone, insulin, melatonin rhythms. There are several issues with the background on chronobiology literature, which weakens the review, as pointed bellow.
Major points>
Line 53: SCN does not work as a feedback loop…
Line 61 and 62> What is the meaning of intake of solar light through optic channels?
Line 65, figure 5: The location of SCN is wrong. By the picture looks like a pituitary gland.
Line 89: Circadian rhythms may also impact contractile alterations …
As in this line, in several paragraphs the authors mistaken using circadian rhythm as a regulator, while the right way to go would be described the function of circadian time systems. Circadian rhythms are output from the circadian time systems, which at the cellular levels rely on the set of transcription factors, named clock proteins. These proteins are interconnected in a feedback loop of transcription/translation that oscillates with a period of 24 h. This loop though with tissue-specificity orchestrates the circadian rhythm of gene expression regulating several cell functions. In mammals, there has been proposed a hierarchical organization, which the central pacemaker is considered the SCN that entrains peripheral oscillators through neural, humoral, and hormonal signals.
Line 93: What is exactly the theory of intrinsic properties of activated muscle?
Line 110: How does the influence of circadian rhythms” on neuromuscular function and muscle strength correlate with the rhythm of gonadal steroids? To this review a connection of this statement should be done with the following paragraphs about the how performance varies according to rhythms of testosterone and cortisol.
Line 126: This diurnal variation in performance…
How this would be explained by muscle circadian clock as there is a temperature compensated mechanism for the intrinsic clocks (see 10.1371/journal.pbio.3002164 ).
Line 287: Pinealocytes is wrong spell it out.
Line 290: Acting as a natural rhythm synchronizer, melatonin…
In this paragraph a reference is missing.
Line 311: Hormonal feedback loop
This entire topic should be revised because there are several issues with the idea of feedback loop between the periphery and the SCN, the central clock. Actually, the dominant idea in the field is that as a master clock, there are fews (if any) temporal cues from the organism that signal to the SCN (see 10.1007/978-3-642-25950-0_1 ).
Author Response
Comment 1: The review by Augsburger et al rise the problematic about time-of/day variations in sport exercise performance. The subject is relevant for a broad audience as there are few evidence on the mechanism by the circadian time system can determine performance. The authors prepared a good table to correlate the time of exercise, recording of core body temperature and performance. Nevertheless, this table is poorly explored in the arguments throughout the review, which shallow touch the mechanism behind the correlation of circadian rhythm and performance. They should also bring a table like that to correlate other circadian rhythm such as cortisol, testosterone, insulin, melatonin rhythms. There are several issues with the background on chronobiology literature, which weakens the review, as pointed bellow.
Response 1: Thank you so much for your review of our manuscript and for your suggestions! We hope that we have addressed all of your concerns in the point-by-point responses presented hereafter. We have also improved Table 1 per your recommendation (line 160). However, because there are so few original human research trials investigating the moderating role of the time-of-day hormonal fluctuations on sport and exercise performance, the additional table that you have requested cannot be implemented.
Comment 2: Line 53: SCN does not work as a feedback loop…
Response 2: Thank you so much for catching this! We have corrected this section in accordance with your recommendations (lines 57-59).
Response 3: Line 61 and 62> What is the meaning of intake of solar light through optic channels?
Response 3: Thank you so much for your question. Simply, we mean to say, “receiving sunlight into the eyes”. We have adapted the sentence based on your recommendation so that this is clearer to the readers (line 62).
Comment 4: Line 65, figure 5: The location of SCN is wrong. The picture looks like a pituitary gland.
Response 4: Thank you so much for this suggestion! Based on your recommendation and the recommendations of the other reviewers, we have omitted Figure 1 from the manuscript.
Comment 5: Line 89: Circadian rhythms may also impact contractile alterations … As in this line, in several paragraphs the authors mistaken using circadian rhythm as a regulator, while the right way to go would be described the function of circadian time systems. Circadian rhythms are output from the circadian time systems, which at the cellular levels rely on the set of transcription factors, named clock proteins. These proteins are interconnected in a feedback loop of transcription/translation that oscillates with a period of 24 h. This loop though with tissue-specificity orchestrates the circadian rhythm of gene expression regulating several cell functions. In mammals, there has been proposed a hierarchical organization, which the central pacemaker is considered the SCN that entrains peripheral oscillators through neural, humoral, and hormonal signals.
Response 5: Thank you so much for your suggestion and insight! Based on this comment and the comments made by other reviewers, we have corrected the term “rhythm” to “regulation”, when appropriate, throughout the manuscript. We have also corrected the description of the circadian time systems based on your recommendation (lines 53-65).
Comment 6: Line 93: What is exactly the theory of intrinsic properties of activated muscle?
Response 6: Thank you so much for your question! We have rephrased this sentence as requested so that these properties are stated explicitly (lines 74-78).
Comment 7: Line 110: How does the influence of circadian rhythms” on neuromuscular function and muscle strength correlate with the rhythm of gonadal steroids? To this review a connection of this statement should be done with the following paragraphs about the how performance varies according to rhythms of testosterone and cortisol.
Response 7: Thank you so much for your question and comments. We have included additional information into this sentence as requested, and have provided an additional connection to the following paragraphs starting with section 3.2 (lines 94-101).
Comment 8: Line 126: This diurnal variation in performance… How this would be explained by muscle circadian clock as there is a temperature compensated mechanism for the intrinsic clocks (see 10.1371/journal.pbio.3002164 ).
Response 8: Thank you so much for your question. We have noted in the manuscript that these circadian shifts in body temperature occur within a physiological range to note that the system is somewhat resilient (in rodent models, per your reference) to external temperature changes (line 113). However, we did not state that variations in performance were directly due to a “muscle circadian clock”, and instead stated that circadian fluctuations in core body temperature indirectly result in the heating of muscles that can influence performance. Moreover, your reference states that the temperature compensation property of circadian systems operate within a physiological range, where small changes within that range can have wide ranging effects on performance. As such, we have selected not to alter this section any further to avoid confusion.
Comment 9: Line 287: Pinealocytes is wrong spell it out.
Response 9: Thank you so much for catching this! We have corrected this as requested (line 274).
Comment 10: Line 290: Acting as a natural rhythm synchronizer, melatonin… In this paragraph a reference is missing.
Response 10: Thank you so much for catching this! We have added references as requested (line 280).
Comment 11: Line 311: Hormonal feedback loop. This entire topic should be revised because there are several issues with the idea of feedback loop between the periphery and the SCN, the central clock. Actually, the dominant idea in the field is that as a master clock, there are fews (if any) temporal cues from the organism that signal to the SCN (see 10.1007/978-3-642-25950-0_1 ).
Response 11: Thank you so much for your suggestion. We have revised this paragraph in accordance with your position and have removed information regarding the feedback loop between the SCN and the periphery (lines 308-370).
Reviewer 2 Report
Comments and Suggestions for Authors
In figure 1 what is marked as SCN is actually pituitary gland. SCN is located in the anterior part of the hypothalamus. Please see the link below. https://en.wikipedia.org/wiki/Suprachiasmatic_nucleus#/media/File:Suprachiasmatic_Nucleus.jpg
How does chronotype affect the diurnal pattern of hormonal fluctuation discussed in this article or body temperature? Please add a paragraph discussing the hormonal fluctuation profile in different chronotypes.
Authors have used the term sport performance in generality. I was wondering if different facets of sport performance are affected similarly by each of the factors considered in this study. For example, fine motor control that is needed in technical skill-based sports like soccer or tennis, muscle power and anaerobic capacity. Consequently, if there is a difference on how these different aspects of sport performance are influenced by these factors (hormonal fluctuations, body temperature), are different types of sports or even positions in the same sport (e.g., goaler vs midfielder in soccer) should be treated differently based on how heavily they rely on fine motor control vs muscle power or anaerobic capacity.
Authors indicate a need for time separation between the end of exercise and start of sleep, I was wondering how long before desired bedtime should the exercises be terminated. Please elaborate further.
Authors indicate the importance of consideration for individual hormonal profile and chronotype in optimizing sport performance. I was wondering if authors have any recommendation on how these aspects could be addressed in a team sport where individual chronotypes and hormonal profile could vary between individual members of the same team.
Could you add a discussion regarding the role of other external factors important in circadian entrainment such as food or possibly taking melatonin pills?
Author Response
Comment 1: In figure 1 what is marked as SCN is actually pituitary gland. SCN is located in the anterior part of the hypothalamus. Please see the link below. https://en.wikipedia.org/wiki/Suprachiasmatic_nucleus#/media/File:Suprachiasmatic_Nucleus.jpg
Response 1: Thank you so much for your review of our manuscript and for this suggestion! Based on your recommendation and the recommendations of the other reviewers, we have omitted Figure 1 from the manuscript.
Comment 2: How does chronotype affect the diurnal pattern of hormonal fluctuation discussed in this article or body temperature? Please add a paragraph discussing the hormonal fluctuation profile in different chronotypes.
Response 2: Thank you so much for your question! In reference to this comment, a new paragraph has been added describing the hormonal fluctuations of the different chronotypes (lines 402-413).
Comment 3: Authors have used the term sport performance in generality. I was wondering if different facets of sport performance are affected similarly by each of the factors considered in this study. For example, fine motor control that is needed in technical skill-based sports like soccer or tennis, muscle power and anaerobic capacity. Consequently, if there is a difference on how these different aspects of sport performance are influenced by these factors (hormonal fluctuations, body temperature), are different types of sports or even positions in the same sport (e.g., goaler vs midfielder in soccer) should be treated differently based on how heavily they rely on fine motor control vs muscle power or anaerobic capacity.
Response 3: Thank you so much for your comment. Although a detailed discussion of how different facets of sport and exercise performance are affected by each of the factors presented is beyond the scope of this review, we have added a section to the manuscript acknowledging that these terms are used generally, and that coaches and practitioners should consider these variations as part of a holistic training approach, and that future research is warranted to explore these differences further (lines 509-517).
Comment 4: Authors indicate a need for time separation between the end of exercise and start of sleep, I was wondering how long before desired bedtime should the exercises be terminated. Please elaborate further.
Response 4: Thank you so much for your question! We have elaborated on the recommended time separation between the end of exercise and bedtime. A detailed explanation, including the suggested 2-3 hour window, has been incorporated into the revised manuscript (lines 297-306).
Comment 5: Authors indicate the importance of consideration for individual hormonal profile and chronotype in optimizing sport performance. I was wondering if authors have any recommendation on how these aspects could be addressed in a team sport where individual chronotypes and hormonal profile could vary between individual members of the same team.
Response 5: Thank you so much for your comment! We have addressed the consideration of individual chronotypes and hormonal profiles within team sports by suggesting tailored approaches, such as chronotype assessments and flexible training schedules, to optimize performance while accounting for individual variability among team members (lines 457-468).
Comment 6: Could you add a discussion regarding the role of other external factors important in circadian entrainment such as food or possibly taking melatonin pills?
Response 6: Thank you so much for your suggestion! We have included a discussion on additional external factors that play a role in circadian entrainment, particularly food timing and melatonin supplementation (lines 420-425).